# Development and Characterization of High-Energy Protein Bars with Enhanced Antioxidant, Chemical, Nutritional, Physical, and Sensory Properties

**DOI:** 10.3390/foods13020259

**Published:** 2024-01-13

**Authors:** Rawan AlJaloudi, Maher M. Al-Dabbas, Hani J. Hamad, Rawan A. Amara, Zaher Al-Bashabsheh, Mahmoud Abughoush, Imranul H. Choudhury, Bha’a Aldin Al-Nawasrah, Sehar Iqbal

**Affiliations:** 1Zarqa University College, Al-Balqa Applied University, Zarqa 13313, Jordan; rawan.aljaludi@bau.edu.jo; 2Department of Nutrition and Food Technology, Faculty of Agriculture, The University of Jordan, Amman 11942, Jordan; m.aldabbas@ju.edu.jo (M.M.A.-D.); rawanamara.ra@gmail.com (R.A.A.); dr.bhaa.alnawasrah@icloud.com (B.A.A.-N.); 3Nutrition and Dietetics Program, College of Pharmacy, Al Ain University, Abu Dhabi 112612, United Arab Emirates; mahmoud.abughoush@aau.ac.ae (M.A.); imranul.haq@aau.ac.ae (I.H.C.); 4Department of Clinical Nutrition and Dietetics, Faculty of Allied Medical Sciences, Philadelphia University, Amman 19392, Jordanzalbashabsheh@philadelphia.edu.jo (Z.A.-B.); 5Department of Clinical Nutrition and Dietetics, Faculty of Applied Medical Sciences, The Hashemite University, Zarqa 13133, Jordan; 6AAU Health and Biomedical Research Center, Al Ain University, Abu Dhabi 112612, United Arab Emirates

**Keywords:** protein bars, proximate analysis, DPPH, phenolic content, flavonoid content, sensory analysis

## Abstract

Protein-rich energy bars are known as an excellent nutritional supplement for athletes that help to build and repair connective tissues. The study is, therefore, aimed at developing high-protein bars using lupine seeds, wheat germ, and selected dried fruits including raisins, dates, apricots, and cranberries. Different formulations (F1, F2, F3, and F4) were performed at different ratios of ingredients to produce high-protein bars and compared them with a control bar made of whey-protein concentrate and oat flakes. For this purpose, a proximate analysis, total phenol content, total flavonoid content, DPPH radical scavenging activity, water content, nutritional, and sensory analysis was performed to evaluate the results. The proximate analysis of the produced protein bars showed a significantly higher protein content (22 ± 2) and total phenolic activity (57 ± 33) in formulation group 4 as compared to the other groups. Furthermore, the least water activity content was found in formulation group 1 (1 ± 0.0) when compared with the control group (1 ± 0.0). The results from the sensory evaluation revealed that T3 had the highest average scores in overall consumer acceptability. Our study found that total phenolic, flavonoid, and fiber content were significantly higher in the prepared protein bars indicating prospective health benefits when compared to the control group. Overall, the study demonstrates that high-protein bars using functional ingredients like dried fruit can provide enriched nutritionally valuable food options for consumers.

## 1. Introduction 

Recent sport research has focused on the formulation of different nutritional bars due to increased consumer demand, high nutritional value, and quick energy availability for the body’s metabolic activities, and considering that protein-rich energy bars are known as an excellent nutritional supplement for athletes that helps to build and repair connective tissue [1]. Apparently, protein bars contain more than 20 g of protein per serving are fortified with vitamins, minerals, and antioxidants while having low carbohydrate and sodium content. Different nutritional bars with ingredients such as cereals, oats, kidney beans, raisins, dates, bananas, coconut flakes, and many more have already proved to be beneficial for athletes [2].

Adding to that, previous studies have reported the anti-aging, anti-chronic disease, and proliferative effects of wheat germ [3]. Wheat germ is also known as an ergogenic aid due to its vitamin-E-rich content that can improve athlete performance [4]. Studies reported that wheat germ has beneficial physiological effects in maintaining normal cholesterol levels and reducing pathogenic gastrointestinal microflora [5]. Also, different epidemiological studies have suggested that replacing several meat meals per week with legumes (lupine) has a positive impact on longevity, diabetes, cardiovascular disease, and weight management due to its potential beneficial effects on the gut microbiome [6]. Lupine appears to consist of about 34% protein and is rich in natural antioxidant capacity and considered one of the best sources of plant protein [7].

Lupine seeds and wheat germ are rich plant-based sustainable proteins with high nutrient density, low allergenicity, and can be excellent options for individuals following a vegan diet. Selecting lupine seeds and wheat germ as a food source or making a protein bar can add several novel, sustainable, and beneficial health and nutritional aspects to the product. Lupine seeds are generally well-tolerated, and their protein content can be a suitable alternative for people with digestive issues. In concomitant, wheat germ is a well-recognized nutritious food rich in vitamins, minerals, and antioxidants. It contains B vitamins, vitamin E, and essential minerals like zinc and magnesium which makes it a suitable ingredient to improve the nutritional profile of protein bars [8]. Moreover, its nutty flavor might help to enhance the flavor and sensory experience for the consumer, as well as the overall acceptability of the product.

The recent shift to urbanization, the modern lifestyle, and long working days with changes in eating habits have shifted traditional meals to quicker, easy to grab but nutritious food. High-protein bars, in this regard, can be considered a quick snack to satisfy temporary hunger, enhance muscle growth, and fulfill nutritional needs. Therefore, this study is specifically designed to produce nutritious, high-quality protein-rich bars from lupine seeds and wheat germ. Additionally, other health additives (dried fruits and sesame) were added along with lupine seeds and wheat germ to enhance its nutritional value, palatability, antioxidant properties, and sensory quality.

The objective of the current study is to formulate and characterize high-protein bars using lupine seeds, wheat germ, and selected dried fruits, including raisins, apricots, dates, cranberries, sesame, and peanut butter. The newly developed bars are expected to enhance the phytochemical, nutritive, and sensory properties, as well as the shelf life, of the product. Also, it is important that the finished product satisfies consumer’s desire for convenience and health advantages; therefore, a comparative analysis of the chemical and sensory quality of the raw material and final products was also performed to meet precise nutritional requirements and for future product status as the best-suited nutritional supplement for athletes.

## 2. Materials and Methods

### 2.1. Chemicals and Ingredients of Protein Bars

Gallic acid, rutin, aluminum trichloride (AlCl_3_), Folin-ciocalteu, and DPPH (1,1-diphenyl-2-picrylhydrazyl) were of analytical grade and were bought from Sigma-Aldrich (Steinheim, Germany). Sodium hydroxide (NaOH), sulfuric acid (H_2_SO_4_), boric acid (H_3_BO_3_), potassium sulfate (K_2_SO_4_), copper (II) sulfate (CuSO_4_), sodium carbonate (Na₂CO₃), diethyl ether (C_2_H_5_)_2_O), and ethanol were of reagent grade and were purchased from local companies in Amman-Jordan. Lupine (China), wheat germ (local wheat-milling company, Amman, Jordan), dates (Saudi Arabia), dried fruit (raisins, apricots, and cranberries from China), coconut oil (Indonesia), peanut butter (China), sesame (Sudan), and xanthan gum (China) as a thickening agent were procured from a local market of Amman-Jordan. The Lupine seeds and wheat germ were proximately analyzed, while USDA tables were used to determine the composition of other raw ingredients. To prepare control bars, we used concentrated whey protein, coconut flakes, and oats, also purchased from the Amman (Jordan) local market.

### 2.2. Preparation of Protein Bars

The experimental bars were prepared using Szydłowska et al.’s (2020) method [9], through combining and mixing all ingredients (Table 1). After combining all the prepared ingredients of the recipe, the homogeneous mass was stored in a refrigerator at a temperature of 4 C for 10 h and used as a control sample. For preparation of the high-protein bar formulations (F1, F2, F3, and F4), lupine seeds were boiled for 10 min and crushed using a blender. High-protein bars were made from crushed parboiled lupine, and then they were mixed with roasted wheat germ. The amount of lupine and wheat germ was randomly and gradually added to increase the protein content from lupine and wheat germ to ensure good structure and adhesion of the final product for each formulation. All ingredients including dried fruits were crushed, combined, mixed, and the final product was then shaped into bars. Shaped bars were stored at a temperature of 4 °C for 10 h. Different functional products and compositions to make multiple formulations of protein bars are shown in Table 1. Later, the protein bars from each formulation were used for chemical and sensory analysis.

### 2.3. The Nutritional Value and Production Cost of Protein Bars

The nutritive value was calculated based on the proportion of ingredients used in the protein bars using the USDA tables for food composition [10], while the cost of the protein bars (fills/100 g) was determined considering the cost of the raw materials (local market unit price) used to produce different formulations of bars.

### 2.4. Chemical Analysis

#### 2.4.1. Proximate Composition of the Protein Bars

The proximate composition (moisture, ash, protein, fat, and crude fiber) of the formulations and control was carried out using the standard methods of AOAC [11]. All analyses were conducted in triplicate. Briefly, moisture was determined gravimetrically using an oven (Memmert, Model UFE500, Schwabach, Germany). Ash determination was accomplished by incinerating 2.0 g of the sample in triplicate, in a muffle furnace at 600 °C, for 20–24 h, using a porcelain crucible. Crude-protein determination was accomplished following the Kjeldahl method using a nitrogen-conversion factor of 6.25 to find the approximate value of total protein content present in the formulation. Crude fat was determined using the Soxhlet extraction method using diethyl-ether as the solvent. The crude fiber was determined after digestion of samples (0.8 g) with dilute acid followed by a diluted base using the digestion method.

#### 2.4.2. Water Activity

The water activity (a_w_) of each formulation and control was determined using an electrical thermoconstanter (Novasina, RTD-200, Lachen, Switzerland) hygrometer at 16 ± 1 °C [12], after calibration against the standards of known saturated salt solutions, which included NaBr, NaCl, (NH_4_)_2_SO_4_, KCl, BaCl_2_, and K_2_SO_4_. Reliable aw values were obtained after hygroscopic equilibrium according to the instruction manual.

### 2.5. Sample Preparation for Analysis

To analyze the total phenolic and flavonoid content and DPPH free-radical-scavenging activity, the ethanol extract 10% (*w*/*v*) from the protein bars, lupine, and wheat germ were prepared separately. Briefly, 10.0 g of ground samples from each formulation or raw ingredients were macerated in 50 mL of absolute ethanol for 15 min, and then the filtrate of each sample was diluted to 100 mL in a volumetric flask with absolute ethanol.

### 2.6. Determination of Total Phenolic Content

The total phenolic content present in the formulated bar was determined using the Folin–Ciocalteau reagent [13]. For this purpose, 1.0 mL of the bar’s ethanol extract (100 mg) was transferred into a 10 mL volumetric flask, followed by the addition of 2.5 mL of distilled water. The Folin–Ciocalteu reagent (250 µL) was then added, followed by mixing thoroughly. After 3 min, 0.5 mL of 10% sodium carbonate (10 g/100 mL) was added, and the absorbance was measured at 760 nm, with a Uv-Vis double beam spectrophotometer (model UVD-2900, Labomed, Los Angeles, CA, USA). Gallic acid was used as the standard for a calibration curve. The total phenolic compound contents (mg/100 g) were expressed as the Gallic acid equivalent and determined from the following regression equation based on the established calibration curve.
y = 0.0408x   R^2^ = 0.9972
where y is the absorbance and x is the Gallic acid concentration in ppm; all measurements were carried out in triplicate.

### 2.7. Determination of Total Flavonoid Content

The flavonoid content was determined using Miliauskas’s method (2004) [14]. Briefly, 1.0 mL from each bar formulation was mixed with 1 mL of 2% aluminum trichloride in an ethanol solution; the mixture was diluted with water into 25 mL and allowed to stand for 40 min at 20 °C, and then the absorption at 415 nm was recorded with a Uv-Vis double-beam spectrophotometer (model UVD-2900, Labomed, Los Angeles, CA, USA). The total flavonoid content (mg/100 g) was expressed as the Rutin equivalent and determined from the following regression equation based on an established calibration curve.
y = 0.0295x − 0.0297   R^2^ = 0.9942
where y is the absorbance and x is the Rutin concentration in ppm; all measurements were completed in triplicate.

### 2.8. Determination of Antioxidant Activities

#### DPPH Free-Radical-Scavenging Assay

DPPH (1,1-diphenyl-2-picrylhydrazyl) was used to determine the free-radical-scavenging activity in the bar using the method of Hatano [15]. In this regard, 1.0 mL of the ethanol extract of each bar formulation (100 mg) was mixed using a vortex with 3 mL of an ethanolic solution of DPPH (6 × 10^−5^ M). The absorbance was measured at 517 nm with a Uv-Vis double-beam spectrophotometer (model UVD-2900, Labomed, Los Angeles, CA, USA) after 30 min against a blank prepared from similar concentrations.
DPPH free-radical-inhibition activity (%) = Control absorbance − (Sample absorbance − Blank absorbance)/Control absorbance × 100

### 2.9. Hedonic Evaluation

A hedonic evaluation of the modified bar was conducted in the Food Science Laboratories at the University of Jordan. Consumers including students and staff (n = 35) were asked to quantify the following quality attributes: (i) overall acceptability, (ii) appearance, (iii) taste, (iv) aroma, and (v) texture. A 9-point hedonic scale (1 = dislike extremely to 9 = like extremely) according to Meilgaard et al.’s study was used to rank the acceptability of the samples [16]. An informed consent form was signed by all the participants for the sensory evaluation. Samples were coded using random three-digit numbers in a randomized serving order. The study was conducted in accordance with the Declaration of Helsinki and approved by the Institutional Review Board of University of Jordan (Jordan) (10 December 21). 

### 2.10. Statistical Analysis

Statistical calculations were performed using the statistical analysis system, SAS program, 2000 (SAS Institute Inc., Cary, NC, USA). Significant differences among means of formulation were determined using the LSD test. Differences at *p* < 0.05 were considered significant. Regression equations and correlation coefficients (r) were determined using MS Excel software (2016). All analyses for formulations were conducted in triplicate.

## 3. Results and Discussion

### 3.1. Composition of the Protein Bars

The results of the proximate composition of all formulations of protein bars (F1, F2, F3, F4, and control), with lupine seeds and wheat germ are shown in Table 2. Lupine seeds independently have shown a high content of crude protein (32.25%) and crude fiber (18.9%), low content of crude fat (2.54%), ash (0.97%), and low calories (344.85 Kcal/100 g), while wheat germ had the highest content of protein (36.66%), crude fiber (18.2%), crude fat (11.16%), ash (3.94%), and calories (409 Kcal/100 g).

The moisture content in formulation groups ranged from 24.88 to 30.38%, while the moisture content of the control bar was 25.74%. F4 showed a significantly higher moisture content as compared to other formulations and the control group (*p* < 0.05). This can be attributed to the differences in the concentration of lupine (25 g) in F4 containing the highest moisture content shown in Table 2. The ash content ranged from 1.77% to 2.28% for the s groups, whereas the ash content of the control bar was 1.98%. Considering that, F1 contains the highest amount of ash and total inorganic residue as compared to other formulation bars and control groups. This change might be due to it containing the highest amount of wheat germ at 21.4 g which was reported with high amount of ash (3.94%), as shown in Table 2. Similar to our results, a previous study reported the highest amount of ash in wheat germ [8].

In the results, we found the amount of protein ranging from 18.70% to 22.40% in the formulation groups, and 30.49% was reported in the control group. The control group protein content was significantly different from other formulations. Also, higher protein content was reported in F1 and F4 as compared to F2 and F3. Changes in protein content of different formulations might be explained due to the difference of lupine and wheat germ content, as F1 contains the highest amount of lupine and the lowest amount of wheat germ, whereas F4 contains the highest amount of wheat germ and the lowest amount of lupine (Table 2). Furthermore, the differences in the amount of protein added (Table 1) in relation to the amount of obtained protein in the final products (Table 2) may be due to the addition of other rich protein ingredients (dry fruits), production methods (grinding or soaking in hot water), as well as the efficiency of production. Our results agreed with the study of Szydłowska et al. (2020) [9].

In the investigated bars, the highest amount of fat compounds (18.97%) was reported in the F3, whereas the control has the lowest amount of fat (8.57%), which may be attributed to the presence of oats in the control group, which have the lowest fat content. Fiber plays an important role in the prevention of obesity, diabetes, and cardiovascular disease. Furthermore, incorporating protein and fiber-rich sources into the diet will promote fullness and satisfaction while also delivering the important nutrients suggested for daily intake. Protein-rich ingredients in bars include oats, peanuts, soybean flour, soy flour, and other types of legumes [9]. Results presented in Table 2 show the average fiber contents of the protein bars ranged from 5.79 (control group) to 10.48% in the F4 group. There were no significant differences (*p* < 0.05) observed between all the protein-bar formulations; however, the fiber content of the control bar was significantly different to the other formulations.

Regarding the nitrogen-free extract (NFE), it comprises mostly soluble sugar, starches, and minute amounts of other components. NFE contents ranged from 17.49 to 23.89 in the formulation groups, while being 27.42% in the control group. Considering that, the control group showed a significant difference (*p* < 0.05) as compared to the formulation groups. This can be attributed to the differences in the concentration of the functional ingredients and from the addition of more lupine and wheat germ which increases the lupine content and decreases the NFE content.

Meal-replacement products should deliver approximately 300 calories per serving, 8 to 10 g of protein (25–50% of total energy in a product), and 100% of the RDA for at least 12 key vitamins and minerals [9]. In our study, the Kcal contents ranged from 308.79 to 341.53. The Kcal content of the control bar was 308.79, which was significantly different to other formulations in the protein-bar group (*p* < 0.05). This can be attributed to the differences in the concentration of the functional ingredients from the addition of lupine and wheat germ which increases the lupine content and decreases the Kcal content.

### 3.2. Water Activity

Results presented in Table 3 show the average water activity (a_w_) contents of the protein bars. The a_w_ contents ranged from 0.783 (F1) to 0.931 for the control group. The highest amount was reported in the control bar and was considered to be optimal for all microorganism’s growth, favoring their multiplication, as previous studies reported that *C. botulinum* type A and B spores can germinate, develop, and produce toxins (0.93 to 0.94), *Salmonella, Clostridium perfringens,* and *Bacillus cereus*. Moreover, the lower limit for mycotoxigenic mold development has been reported to be 0.78 a_w_, and mycotoxin production is generally greater than the minimal values for growth [17]. This creates a natural safety margin for all of our formulation groups.

Water activity is considered one of the prime factors affecting microbial growth, food stability, shelf life, and food toxicity. Water activity along with temperature, oxygen availability, nutritional composition, PH, acidity, and the addition of natural, or included inhibitors is mainly responsible for the inhibition of microorganisms and for minimizing their effects on food composition and rancidity [12].

### 3.3. Total Phenolic and Flavonoid Content

We found a correlation between total phenolic content (TPC) and total flavonoid content (TFC). A TFC value was reported in F4 (56.63% mg GAE/100 g), while TPC was the highest in F1 (18.81% mg RE/100 g). F4 showed the highest phenolic content because of its highest wheat germ content as compared to other protein bars. However, the control protein bar had the lowest phenolic content (13.59 mg GAE/100 g), since it does not contain any of the former functional ingredients. The flavonoid content varied from 5.56 mg RE/100 g in the control to 18.81 mg RE/100 g in F1. The total flavonoid content of the functional ingredients used to produce protein bars from lupine and wheat germ was 9.48 and 20.94 mg RE/100 g, respectively. The highest flavonoid content value was for wheat germ, followed by lupine. Studies showed that lupine and wheat germ contain different amounts of flavonoid content. Oomah et al. (2006) reported that lupine flavonoid content’s range was 4.15–4.95 mg rutin equivalent (RE/g) [18]. Others found a range of 133–1100 µg catechin/g [19]. In 2015, Zou et al. suggested that the total flavonoid content of wheat germ ranged from 15.80 to 15.95 mg schaftoside equivalents/g [20].

Polyphenols are naturally occurring, non-nutritive compounds present in fruits, vegetables, herbs, and plants. Flavonoids are thought to enhance health via a range of cell signaling pathways and antioxidant actions [21].

Phenolic and flavonoid compounds have very robust antioxidant activity that prevents degenerative diseases and retards aging factors [22]. Flavonoids are present in different food items including fruits, vegetables, and tea, and they can be differentiated according to their bioavailability, metabolism, and biological activity based on their chemical structure, number of hydroxyls, and functional group. Previous studies demonstrated the preventive effects of flavonoids on chronic diseases, CVD risks, neurological disorders, and certain cancers [23]. Similarly, most phenolic compounds are thought to provide health benefits, such as lowering the risk of cardiovascular and neurological disease, as well as lowering the risk of cancer, diabetes, and osteoporosis [5]. Studies show that dry apricots, raisins, and dry cranberries contain different amounts of phenolic content according to dry apricot, raisin, and cranberry types. For example, Meng et al. (2011) found that the phenolic content of different types of raisins ranged from 193.3 to 678.4 mg GAE/100 g [24]. The total phenolic content of dry apricots ranged from 839 to 890 mg GAE/100 g [25], while the total phenolic content of dry cranberries ranged 507–709 mg GAE/100 g [26]. Both TFC and TPC activity were significantly higher in our formulation groups as compared to the control group (Table 4) due to the high content of functional food present in the protein bars that contain a high amount of TPC and TFC.

### 3.4. DPPH Radical Scavenging Activity

One DPPH inhibition technique is a concentration-dependent assay. As the concentration of total antioxidant activity increases, more DPPH free-radical-scavenging activity will occur, and lower values will be obtained. This action is accompanied by decolorization as an indicator for DPPH quenching [27]. The DPPH radical-scavenging activities of the protein bars and/or the ingredients to make them are represented in Table 4.

The highest antioxidant activity referred to has the highest DPPH inhibition (%) which was observed in F1 (50.47%), while the control sample has the lowest DPPH inhibition (9.21%) (*p* < 0.05). The antioxidant abilities of the produced bars are directly proportional to the added dried fruits and wheat germ. The DPPH antioxidant activity of different protein bars’ formulation is closely related to the high percentage of date fruit (14 g) and wheat germ and their phenolic contents. The date fruits were previously reported with a high number of phenolic compounds ranging from 381 to 3541 mg GAE/100 g according to the variety or solvent used for extraction [28], while wheat germ used in protein bars of this study contain 33.5 mg GAE/100 g.

Phenolics are a wide group of compounds that have the ability to scavenge certain radicals, and they are mainly from plant sources [29]. However, the activity of antioxidant compounds in plants is highly affected by various factors, such as temperature, heat, processing parameters, solvent polarity, particle size, etc. [30]. In the present study, the F1 bar exhibited the highest DPPH radical-scavenging activity among the formulations. This heightened antioxidant activity can be attributed to its elevated phenolic compound content (56.3 mg GAE/100 g), due to the substantial utilization of wheat germ (21.4 g), while other formulations were ranked in the following decreasing order F1 > F2 > F3 > F4. In this study, a reduction in DPPH antioxidant activity was observed in the control group when compared to the other formulations (F1–F4). This can be attributed to the incorporation of functional ingredients that are rich in phenolic compounds, such as dried dates, wheat germ, raisins, and dried cranberries, within the formulations.

### 3.5. Vitamin and Minerals Profiles and Cost of Protein Bars

The vitamin and mineral profiles and cost of the traditional and developed protein bars were calculated separately concerning the proportion of added ingredients (Table 5). We found a high concentration for most minerals in the formulated protein bars when compared with the control group, while vitamin C, niacin, riboflavin, vitamin B12, vitamin A and E concentration was high in the control group due to composition fractions.

In the control bars, the main ingredients used to formulate the protein bar were whey protein concentrate (WPC) (15.6 g), oat flakes (21.4 g), raisins (10 g), and coconut flakes (4 g). While lupine seeds (15.6–25 g), wheat germ (21.4–12 g), dried fruits (apricot (5 g), cranberries (5 g), and raisins (5 g)) were used to make the new differently formulated bars. Whey protein concentrate (WPC) and oatmeal are high in calcium content (548 and 270 mg/100 g, respectively). In contrast, lupine seeds and wheat germ have lower calcium levels (176 and 1 mg Ca/100 g, respectively).

Table 5 shows that the control bars have the highest levels of calcium and sodium, while in the formulated protein bars calcium and sodium levels increased with the increase of lupine-seed levels in the bars F4 > F3 > F2 > F1. In addition, wheat germ and lupine seeds have higher contents of iron (7.0 and 4.36 mg/100 g, respectively), magnesium (290 and 198 mg/100 g, respectively), phosphorus (700 and 440 mg, respectively), zinc (11 and 4.75 mg/100 g), and potassium (3000 and 1010 mg/100 g, respectively).

Vitamin C, niacin, B12, A, and E levels were highest in the control bar due to the use of WPC, oat flakes, coconut flakes, and higher number of raisins compared to the formulated bars. Other vitamins were found to be high in the formulated bars, particularly folate, which is widely present in lupine seeds (355 µg/100 g) and wheat germ (364 µg/100 g). Evidence from different studies reported that high-protein bars are rich in dietary fibers and important micronutrients and can be used as the best meal replacement for athletes and physically active people [31,32]. Moreover, the cost for the control group was higher than the cost of the formulated protein bars, making them more cost effective.

### 3.6. Hedonic Evaluation

Food quality encompasses both sensory properties perceived by the human senses and hidden attributes, such as nutrition and safety, which help describe the quality of the product. The sensory evaluation of high-protein organic bars was performed immediately following the manufacturing procedure. The sensory analysis results are shown in Table 6. The highest appearance score was obtained for F3 (7.62), and the lowest was obtained for control (6.71). The textural scores showed that F2 has the best texture, while the control group was rated as the worst in texture by the consumers. Similarly, F3 had the highest average scores in overall acceptability (7.42), appearance (7.62), and taste and aroma (7.14).

## 4. Conclusions

The results of this study have shown that high-protein bars using lupine seeds, wheat germ, and dried fruits which enhance the nutritive value, antioxidant activity, sensory quality, acceptable cost, and stability. Total phenolic, flavonoid, and fiber content were significantly higher in the developed protein bars indicating higher antioxidant activity and prospective health benefits when compared to the control group. The formulated lupine bars showed higher level of crude fiber (~10.5%) and of most of the minerals when compared to the control group, while vitamin C, niacin, riboflavin, vitamin B12, and vitamin A and E concentration were higher in control group due to their composition fractions. Moreover, all protein bars exhibited lower water activity than the control, indicating more stability of the formulated bars during storage.

Overall, the study demonstrates that the formulated high-protein bars using functional plant-based ingredients like lupine, wheat germ, and dried fruit can provide enriched nutritionally valuable food options for consumers. High protein is necessary for athletes and the active population for muscle regeneration. New and alternative solutions instead of conventional approaches or products might help to fulfill current and future consumer demands. Our study focused on the formulation of protein-rich bars with lupine seeds and wheat germ that can provide cheap protein sources with a high-protein quality, optimal for nutritional and sensory evaluation, and ideal for longer storage.

## Figures and Tables

**Table 1 foods-13-00259-t001:** Composition of protein bars from selected functional ingredients.

Ingredients	Control [9]	Formulation F (1)	Formulation F (2)	Formulation F (3)	Formulation F (4)
	The amount of ingredient (g)				
**whey protein,** **concentrated**	15.6	-	-	-	-
**coconut flakes**	4	-	-	-	-
**oat flakes**	21.4	-	-	-	-
**lupine seeds**	-	15.6	20	22	25
**wheat germ**	-	21.4	17	15	12
**raisins**	10	5	5	5	5
**dried apricots**	-	5	5	5	5
**dried cranberries**		4	5	5	5
**dried dates**	14	14	14	14	14
**sesame**	7	7	7	7	7
**peanut butter**	14	14	13	13	13
**water**	7	7	7	7	7
**coconut oil**	7	7	7	7	7
**total**	100 g	100 g	100 g	100 g	100 g

**Table 2 foods-13-00259-t002:** Chemical composition and calories of protein-bar formulations and control group.

Sample (Formulation)	Moisture (%)	Ash (%)	Protein (%)	Fat (%)	Fiber (%)	NFE ^1^ (%)	Kcal/100 g
**F1**	24.88 ± 0.93 ^c^	2.28 ± 0.05 ^a^	20.21 ± 1.65 ^bc^	18.33 ± 0.6 ^a^	10.42 ± 0.27 ^a^	23.89 ± 0.90 ^b^	341.53 ± 3.25 ^a^
**F2**	28.08 ± 0.18 ^b^	1.77 ± 0.26 ^b^	19.78 ± 0.01 ^c^	17.48 ± 0.27 ^b^	10.44 ± 0.23 ^a^	22.43 ± 0.15 ^b^	326.23 ± 3.15 ^b^
**F3**	28.47 ± 0.16 ^b^	1.97 ± 0.01 ^b^	18.70 ± 1.64 ^c^	18.97 ± 0.72 ^a^	10.46 ± 0.15 ^a^	21.42 ± 2.30 ^b^	331.26 ± 3.96 ^b^
**F4**	30.38 ± 0.37 ^a^	1.90 ± 0 ^b^	22.40 ± 1.59 ^b^	17.34 ± 0.13 ^b^	10.48 ± 0.32 ^a^	17.49 ± 1.94 ^c^	315.64 ± 1.29 ^c^
**Control**	25.74 ± 0.028 ^c^	1.98 ± 1.98 ^b^	30.49 ± 0.48 ^a^	8.57 ± 0.04 ^c^	5.79 ± 0.47 ^b^	27.42 ± 0.73 ^a^	308.79 ± 1.06 ^d^

Results are meant ± standard deviation of triplicate analyses. Values with different letters are significantly different (*p* < 0.005). ^1^ Nitrogen-free extract.

**Table 3 foods-13-00259-t003:** Average water activity content *.

Formulation	Water Activity
**F1**	0.783 ± 0.005 ^d^
**F2**	0.807 ± 0.002 ^c^
**F3**	0.847 ± 0.002 ^b^
**F4**	0.842 ± 0.002 ^b^
**Control**	0.931 ± 0.005 ^a^
**Lupine**	0.588 ± 0.002
**Wheat germ**	0.847 ± 0.003

* Results mean ± standard deviation of triplicate analyses. Values with different letters are significantly different (*p* < 0.005).

**Table 4 foods-13-00259-t004:** Average phenolic content as gallic acid (mg GAE/100 g), flavonoid content as rutin (mg RE/100 g), and DPPH inhibitory activity (%).

Formulation	Phenolic Content as Gallic Acid (mgGAE/100 g)	Flavonoid Content as Rutin (mg RE/100 g)	DPPH Inhibition (%) (1000 ppm)
**F1**	56.27 ± 4.57 ^a^	18.81 ± 0.82 ^a^	50.47 ± 1.92 ^a^
**F2**	53.57 ± .0.68 ^a^	14.10 ± 0.06 ^b^	40.16 ± 0.28 ^b^
**F3**	46.56 ± 2.45 ^b^	13.61 ± 0.11 ^b^	32.28 ± 0.05 ^d^
**F4**	56.63 ± 32.71 ^a^	12.83 ± 0.60 ^c^	35.34 ± 1.78 ^c^
**Control**	13.59 ± 2.42 ^c^	5.56 ± 0.46 ^d^	9.21 ± 0.55 ^e^
**Lupine**	16.81 ± 4.38	9.48 ± 1.12	8.80 ± 0.19
**Wheat germ**	33.54 ± 1.03	20.94 ± 1.67	51.23 ± 5.57

Results are meant ± standard deviation of triplicate analyses. Values with different letters are significantly different (*p* < 0.005).

**Table 5 foods-13-00259-t005:** Vitamin and mineral profiles of the protein bars, and the unit production cost of 100 g.

Minerals (mg)	F1	F2	F3	F4	Control
Calcium	126.73	133.47	136.21	140.32	212.17
Iron	3.72	3.6	3.58	3.52	2.75
Magnesium	117.72	115.96	115.14	113.91	101.61
Phosphorus	310.30	292.7	284.66	272.6	240.88
Potassium	575.80	581.49	583.85	587.39	418.1
Sodium	7.81	8	8.06	8.15	64.30
Zinc	4.02	3.68	3.53	3.31	2.78
**Vitamins**					
Vitamin C (mg)	0.92	1.13	1.23	1.37	5.67
Thiamin (mg)	0.57	0.51	0.49	0.45	0.30
Niacin (mg)	2.53	2.30	2.20	1.43	3.35
Riboflavin (mg)	0.23	0.16	0.16	0.15	0.22
Vitamin B_6_ (mg)	0.44	0.40	0.38	0.35	0.30
Folate (µg)	125.15	128.41	129.89	132.11	52.58
Vitamin B_12_ (µg)	-	-	-	-	0.54
Vitamin A (µg)	10.06	10.08	10.08	10.08	137.01
Vitamin E (mg)	0.33	0.35	0.35	0.03	1.35
Cost/100 g (fils)	400.6	404.97	404.97	404.97	621.84

**Table 6 foods-13-00259-t006:** Average score of the modified protein bars’ properties *.

Formulation	Overall Acceptability	Appearance	Taste and Aroma	Texture
**F1**	7.05 ± 1.67 ^ab^	7.14 ± 1.78 ^ab^	6.74 ± 1.88 ^a^	7.28 ± 1.46 ^a^
**F2**	7.02 ± 1.50 ^ab^	7.08 ± 1.70 ^ab^	7.31 ± 1.36 ^a^	7.60 ± 1.37 ^a^
**F3**	7.42 ± 1.28 ^a^	7.62 ± 1.30 ^a^	7.14 ± 1.21 ^a^	7.57 ± 1.37 ^a^
**F4**	6.65 ± 1.71 ^b^	7.02 ± 1.61 ^ab^	6.80 ± 1.54 ^a^	7.31 ± 1.49 ^a^
**Control**	5.74 ± 1.82 ^c^	6.71 ± 2.12 ^b^	5.94 ± 1.76 ^b^	6.25 ± 1.96 ^b^

* Results mean ± standard deviation of triplicate analyses. Values with different letters are significantly different (*p* < 0.005).

## Data Availability

The original contributions presented in the study are included in the article, further inquiries can be directed to the corresponding author.

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
