# Peer review of "Development and Characterization of High-Energy Protein Bars with Enhanced Antioxidant, Chemical, Nutritional, Physical, and Sensory Properties"

_foods, 2024, doi:10.3390/foods13020259_

Round 1

Reviewer 1 Report

Comments and Suggestions for Authors

The objective of this study was to formulate and characterize high-protein bars. The results were unsurprising, given that the bars were produced by the authors, and thus, the composition was well-known. In my assessment, the manuscript resembles more of a quality control report than a research study.

On the whole, the study conducted by the authors presents a standard AOAC-based analysis of the prepared bars without introducing any innovative approaches. The research design and methodology are only briefly outlined, lacking a thorough assurance of the reliability of the findings.

Beyond the lack of originality and limited scientific relevance, I believe the manuscript's weakness lies in the description of the samples and the uselessness of the statistical treatment applied to laboratory-made samples using well-known ingredients. The authors should consider revising the manuscript to emphasize the novelty of their approach and the scientific innovation of the experiments, particularly if the manuscript's focus is on the influence of raw materials (such as lupine, wheat germ) rather than the bar production process.

Furthermore, additional information and discussion on sample preparation, the rationale behind performing statistical analyses, and the anticipated results should be included, especially since the samples are self-made. Is the statistical approach employed to verify the accuracy of the preparation protocol?

The following specific points should be addressed:

  1. Significant Figures: Revise throughout the manuscript. For instance, on line 39, "22.40±1.59" should be written as "22±2," and "56.63±32.71" should be written as "57±33," etc.
  2. Control Group (Line 35): Provide a brief description of the control group mentioned.
  3. Subscripts (Line 90): Revise subscripts as necessary.
  4. Chemical Information (Line 92): Include the name of the company and purity grade of the chemicals used in the study.
  5. Raw Materials (Lines 92-96): Clarify whether the raw materials (ingredients) are characterized before use or if they are accompanied by a certificate of analysis.
  6. Szydlowska Method (Line 98): Briefly describe the Szydlowska method, including any modifications.
  7. Term "Treatment" (Line 104): Clarify the term "treatment" and specify what it refers to.
  8. Table 1: Add more details about the "control" bars' production and ingredients.
  9. "Fills/00g" (Line 111): Clarify the meaning of "fills/00g."
  10. Treatment Name (Line 112 and following): If "treatment" refers only to a difference in composition but with the same preparation protocol, explain this better and consider using a different name for the samples.
  11. AOAC in Methods (Line 116): Add "AOAC" between "standard" and "methods" and include "Briefly" before "moisture."
  12. Water Activity (Line 125): Rewrite the "water activity" experiment, adding scientific bases and standard references.
  13. Statistical Analysis (Line 178): Explain the meaning of "All treatments were conducted in triplicate" in the statistical analysis.
  14. Figure 1: Add the meaning of "a" and "b" letters to the caption and include error bars in the graph.
  15. Table 6: Include variability (i.e., standard deviation) in the minerals concentration table to make the results meaningful.

Author Response

Title; Development and Characterization of High Energy Protein Bars with Enhanced Antioxidant, Chemical, Nutritional, Physical and Sensory Properties

Submission ID: foods-2745472

Dear Editor, Dear Reviewers,

We would like to thank the reviewers for their in-depth analysis and suggestions to improve our manuscript. We addressed all points and revised the script accordingly. Revisions are marked in yellow in the script.

We performed revisions especially i) Improved methodology, ii) results and discussion, iii) additional references according to the suggestions. Furthermore, all criticized points were addressed and suggestions from the reviewers were incorporated in the latest manuscript.

We hope that this revision is satisfactory. Thank you.

Reviewer Comments

Author Response

  1. Significant Figures: Revise throughout the manuscript. For instance, on line 39, "22.40±1.59" should be written as "22±2," and "56.63±32.71" should be written as "57±33," etc.

Thank you for your input, we have revised the figures throughout the manuscript and make the changes according to the suggestions.

  1. Control Group (Line 35): Provide a brief description of the control group mentioned.

Description of the control group is now mentioned accordingly.

  1. Subscripts (Line 90): Revise subscripts as necessary.

We have checked and revise the subscripts in the revised manuscript.

  1. Chemical Information (Line 92): Include the name of the company and purity grade of the chemicals used in the study.

Thank you for your in-depth analysis, we have provided the information regarding chemical information as suggested.

  1. Raw Materials (Lines 92-96): Clarify whether the raw materials (ingredients) are characterized before use or if they are accompanied by a certificate of analysis.

Thank you for your input, we have revised the manuscript providing details for raw material. All details are yellow highlighted.

  1. Szydlowska Method (Line 98): Briefly describe the Szydlowska method, including any modifications.

We have modified and provided details for the Szydlowska method accordingly.

  1. Term "Treatment" (Line 104): Clarify the term "treatment" and specify what it refers to.

We have replaced the treatment with formulation for better clarity.

  1. Table 1: Add more details about the "control" bars' production and ingredients.

The suggested details are incorporated in the revised manuscript.

  1. "Fills/00g" (Line 111): Clarify the meaning of "fills/00g."

We apologies for the mistake, it is fills/100g. The explanation is given in the revised manuscript.

  1. Treatment Name (Line 112 and following): If "treatment" refers only to a difference in composition but with the same preparation protocol, explain this better and consider using a different name for the samples.

We have replaced the treatment with formulation for better clarity.

  1. AOAC in Methods (Line 116): Add "AOAC" between "standard" and "methods" and include "Briefly" before "moisture."

We have modified the text according to the suggestions.

  1. Water Activity (Line 125): Rewrite the "water activity" experiment, adding scientific bases and standard references.

The water activity is now revised with more details and explanation and with additional reference.

  1. Statistical Analysis (Line 178): Explain the meaning of "All treatments were conducted in triplicate" in the statistical analysis.

Thank you for your recommendation, we have added the suggested changes accordingly.

  1. Table 6: Include variability (i.e., standard deviation) in the minerals concentration table to make the results meaningful.

Thank you for your valuable suggestion, however it is mentioned that the values given in the table 6 is a result of estimated calculations.

Reviewer 2 Report

Comments and Suggestions for Authors

Dear Authors,

Thank you for your valuable scientific contribution in the manuscript ID: foods-2745472.

The manuscript entitled: Development and Characterization of High Energy Protein Bars with Enhanced Antioxidant, Chemical, Nutritional, Physical and Sensory Properties, by authors: Rawan Al-Jaloudi, Maher M. Al-Dabbas, Hani J. Hamad, Rawan A. Amara, Mahmoud Abughoush, Imranul H. Choudhury, Bha’a Aldin Al-Nawasrah, Zaher Al-Bashabsheh, Sehar Iqbal, is an original article.

My opinion regarding the paper is good.

I consider that it is a valuable article of this topic, which develop  high-protein bars using lupine seeds, wheat germ and selected dried fruits including raisins, date, apricots, and cranberry. Different treatments (T1, T2, T3 and T4) were performed at different ratios and ingredients to produce high protein bars and compared with control group. Representative determinations were performed to evaluate total phenol content, total flavonoid content, DPPH radical scavenging activity, water content, nutritional and sensory analysis. The results from the sensory evaluation revealed that T3 had the highest average scores in overall consumer acceptability. In this study was found that total phenolic, flavonoid, and fiber content were significantly higher in prepared protein bars indicating prospective health benefits when compared to the control group. Overall, the study demonstrates that high protein bars by using functional ingredients like dried fruit can provide enrich nutritionally valuable food options for the consumers.

 I have found a number of minor issues that require attention from the authors.

I marked my suggestions in yellow color in all the manuscript.

I have some suggestions, as follows:

-     Please, there are not clear who are the authors with affiliations 6, 8, 9.

-      Line 90: „(H3BO3)”, I suggest „(H3BO3)”.

-       Line 90: „(K2SO4)” , I suggest „(K2SO4)”.

-   Line 92: Please, mention more clear the name of local companies in Amman-Jordan, which provided the chemicals and reagents used in this study.

-  Line 94: Please, it is possible to explain more specific, the origin of the foods used as raw materials for obtaining the high energy proteins bar; you mention in text procured from a local market of Amman- Jordan”.

I suggest you to insert a blank between the number and the unit of measure, e.g.:

-    Line 102: “4°C”, I suggest4 °C”.

-     Table 1: “100g”, I suggest “100 g”.

-      Line 119: “600°C”, I suggest600 °C”.

-      Lines 143, 153, 190, 191, 192: “(/100g)”, I suggest “(/100 g)”.

-      Line 151: “20°C”, I suggest20 °C”.

-      Line 200: “25g”, I suggest25 g”.

-      Line 205: “21.4g”, I suggest21.4 g”.

-  Lines 259, 260, 262, 263, 264, 266, 269, Table 4, 286, 287, 288, 332): “RE/100g; GAE/100g” I suggest “RE/100 g; GAE/100 g”.

 -            Table 1: “proteinconcentrated”, I suggest “protein concentrated”.

-    Table 1: “Summery”, I suggest “Summary”.

- Line 111: “(fills/00g)”, it is not clear the unit of measure, I suggest “(fills/100 g)”.

-    Line 120: “kjeldahl”, I suggestKjeldahl”.

-    Lines 133, 134, 137, 138, 140, 149, 150, 162, 163: “ml”, I suggest “mL”.

-    Line 139: “μl”, I suggest “μL”.

-   Line 141: “at a 760nm spectrophotometer (model UVD-2900, Labomed, USA)”, I suggest “at 760 nm wavelength, with an UV-Vis Double Beam Spectrophotometer (model UVD-2900, Labomed, USA)”.

- Lines 152-153: “at 415 nm was taken (Labomed spectrophotometer, model UVD-2900, 152 Labomed, USA)”, I suggest “at 415 nm wavelength was taken, with an UV-Vis Double Beam Spectrophotometer (model UVD-2900, Labomed, USA)”.

-   Lines 164-165: “517 nm (Labomed 164 spectrophotometer, model UVD-2900, Labomed, USA)”, I suggest “at 517 nm wavelength, with an UV-Vis Double Beam Spectrophotometer (model UVD-2900, Labomed, USA)”.

-     Line 184: “Result”, I suggest “Results”.

-   Line 248: „C. botulinum”, I suggest Italic font for the name of bacterial strain C. botulinum”.

If the results mentioned in the Table 6. Nutritive value of the protein bars and unit production cost of 100 g, have been already published by the authors, please mention the respective citation, because for minerals and vitamins content, are not described the materials and methods used for analysis, in the present manuscript.

-   At Reference 16, I recommend to use Italic Font for the scientific names of the species, in Latin language.

- At References, according with MDPI requirements, the year of the published article must written in Bold.

-    At Reference 4, the volume and page are not mentioned.

-    At Reference 13, capital letters should not be used.

-   At Reference 15, the identification data regarding the published article is missing (journal, year, volume, pages).

- At References 1, 2 and 17, the pages of the respective articles are missing.

Thank you!

Best regards!

Comments on the Quality of English Language

 Minor editing of English language required.

Author Response

(The authors gave the same response as above.)

Author Response

I suggest you to insert a blank between the number and the unit of measure, e.g.:

Line 90: „(H3BO3)”, I suggest „(H3BO3)” and „(K2SO4)” , I suggest „(K2SO4)”.

Thank you for your recommendation, we have modified the text accordingly.

-   Line 92: Please, mention more clear the name of local companies in Amman-Jordan, which provided the chemicals and reagents used in this study.

-  Line 94: Please, it is possible to explain more specific, the origin of the foods used as raw materials for obtaining the high energy proteins bar; you mention in text “procured from a local market of Amman- Jordan”.

We have provided the more details regarding raw material, chemical and regents according to the suggestions.

I suggest you to insert a blank between the number and the unit of measure, e.g.:

-    Line 102: “4°C”, I suggest “4 °C”.

-     Table 1: “100g”, I suggest “100 g”.

-      Line 119: “600°C”, I suggest “600 °C”.

-      Lines 143, 153, 190, 191, 192: “(/100g)”, I suggest “(/100 g)”.

-      Line 151: “20°C”, I suggest “20 °C”.

-      Line 200: “25g”, I suggest “25 g”.

-      Line 205: “21.4g”, I suggest “21.4 g”.

-  Lines 259, 260, 262, 263, 264, 266, 269, Table 4, 286, 287, 288, 332): “RE/100g; GAE/100g” I suggest “RE/100 g; GAE/100 g”.

Thank you for your in-depth analysis, we have now revised the whole manuscript and modified the suggested text, all changes are yellow highlighted.

-            Table 1: “protein concentrated”, I suggest “protein concentrated”.

-    Table 1: “Summery”, I suggest “Summary”.

Thank you, we have modified the text accordingly.

- Line 111: “(fills/00g)”, it is not clear the unit of measure, I suggest “(fills/100 g)”.

-    Line 120: “kjeldahl”, I suggest “Kjeldahl”.

-    Lines 133, 134, 137, 138, 140, 149, 150, 162, 163: “ml”, I suggest “mL”.

-    Line 139: “μl”, I suggest “μL”.

We apologize for the mistake, it is fills/100 g. we have replaced it with the correct one. Also, the changes are made in line 120, 133, 134, 137, 138, 139, 140, 149, 150, 162, and 163 according to the suggestions.

-   Line 141: “at a 760nm spectrophotometer (model UVD-2900, Labomed, USA)”, I suggest “at 760 nm wavelength, with an UV-Vis Double Beam Spectrophotometer (model UVD-2900, Labomed, USA)”.

- Lines 152-153: “at 415 nm was taken (Labomed spectrophotometer, model UVD-2900, 152 Labomed, USA)”, I suggest “at 415 nm wavelength was taken, with an UV-Vis Double Beam Spectrophotometer (model UVD-2900, Labomed, USA)”.

-   Lines 164-165: “517 nm (Labomed 164 spectrophotometer, model UVD-2900, Labomed, USA)”, I suggest “at 517 nm wavelength, with an UV-Vis Double Beam Spectrophotometer (model UVD-2900, Labomed, USA)”.

We have revised the text according to the recommendations. All changes are yellow highlighted in the revised manuscript.

-     Line 184: “Result”, I suggest “Results”.

-   Line 248: „C. botulinum”, I suggest Italic font for the name of bacterial strain C. botulinum”.

Thank you for your detailed input, we have changed Result to Results. Also, we have now used italic font for C. botulinum

If the results mentioned in Table 6. Nutritive value of the protein bars and unit production cost of 100 g, have been already published by the authors, please mention the respective citation, because for minerals and vitamins content, are not described the materials and methods used for analysis, in the present manuscript.

Thank you for your valuable suggestion, however it was not published before. Similarly, it is mentioned that the values given in table 6 are the results of estimated calculations and not from chemical analysis.

-   At Reference 16, I recommend to use Italic Font for the scientific names of the species, in Latin language.

- At References 1, 2 and 17, the pages of the respective articles are missing.

- At References, according with MDPI requirements, the year of the published article must written in Bold.

-    At Reference 4, the volume and page are not mentioned.

-    At Reference 13, capital letters should not be used.

-   At Reference 15, the identification data regarding the published article is missing (journal, year, volume, pages).

Thank you for your input, we have revised the reference section accordingly.

Reviewer 3 Report

Comments and Suggestions for Authors

It is an interesting study for introduction new functional food, however there are many inconsistencies. Please see some of my comments below:

-Purpose of the study is not explained very well. Besides adding highly nutritious ingredients what is the aim to have 4 different bars? in what way it is different from control? 

-Difference between treatments (better to call it formulations) is not clear, please explain in more detail. 

-Has control been prepared? if yes, why commercial one was not used?

-116-124 can be presented in table 

-Why there is Lupin an wheat germ result in Table 2? have you made bars from only lupin or only wheat germ? Haven't they been ingredients of treatments?

Comments on the Quality of English Language

Need proof reading for language 

Author Response

(The authors gave the same response as above.)

Author Response

It is an interesting study for introduction new functional food, however there are many inconsistencies. Please see some of my comments below:

-Purpose of the study is not explained very well. Besides adding highly nutritious ingredients what is the aim of having 4 different bars? in what way it is different from control? 

Thank you for your positive feedback, we have revised the manuscript according to your valuable suggestions.

We have now revised the study objective and significance. Also, we added explanation regarding control bar and ingredients.

-Difference between treatments (better to call it formulations) is not clear, please explain in more detail. 

Thank you for your suggestion, we have replaced the treatment with formulation for better clarity.

-Has control been prepared? if yes, why commercial one was not used?

We added an explanation regarding control bar preparation; however, we compared our new formulation protein bars with the standard protein bars published by Aleksandra et al., 2020. The details are mentioned in the Table 1.

-116-124 can be presented in table 

Thank you for your input, however the chemical analysis methods are already briefly explained and making a table might create confusion for readers.

-Why there is Lupin an wheat germ result in Table 2? have you made bars from only lupin or only wheat germ? Haven't they been ingredients of treatments?

Thank you, we analyzed Lupin and wheat germ in table 2 because their composition could vary according to the cultivars. Moreover, lupin and wheat germ was used as a main ingredient for standardization of different formulation of protein bars.

Reviewer 4 Report

Comments and Suggestions for Authors

For the improving of the quality of the manuscript it is recommended to add more information in the methodical part 2.2 Preparation of protein bars . How many replicates of protein bars from each treatment were used for chemical and sensory analysis?

Author Response

(The authors gave the same response as above.)

Author Response

For the improving of the quality of the manuscript it is recommended to add more information in the methodical part 2.2 Preparation of protein bars . How many replicates of protein bars from each treatment were used for chemical and sensory analysis?

Thank you for your valuable input, we revised the methodology, chemical and sensory analysis section according to the suggestion. All changes are yellow highlighted in the revised manuscript.

Round 2

Reviewer 3 Report

Comments and Suggestions for Authors

Dear Authors, thank you for considering my feedback. Please, see further comments below:

-How you justify amount of Lupin/Wheat used for each formulation (F1-F4)? Was it based on some standard, was it random etc? As far as I understood referenced method was only for preparation of control bar. You need to include line of sentences where you can explain why you choose certain amount of Lupin and Wheat for each formulation. 

-Table 2- Did you use different Lupin/Wheat cultivar for each formulation? Or it was the same? Please, mention this in methodology. In both cases information about Lupin/Wheat needs to be in separate table (maybe in Appendix, since it is not you main objective)

Comments on the Quality of English Language

Minor grammatical mistakes

Author Response

Dear Reviewer, 

Thank you so much for your positive feedback. We have addressed your both comments in the revised manuscript. All changes are marked yellow in the revised manuscript. 

Reviewer comments

Authors response

-How do you justify the amount of Lupin/Wheat used for each formulation (F1-F4)? Was it based on some standard, was it random etc? As far as I understood the referenced method was only for preparation of control bar. You need to include line of sentences where you can explain why you choose certain amount of Lupin and Wheat for each formulation.

Thank you for your in-depth analysis to improve our manuscript.

Regarding the amount of Lupin/Wheat, the main idea was to raise protein content from lupin and to maintain the texture as well. Therefore, it is randomly used to increase the protein content from lupin and gradually added to ensure good structure and adhesion of the final product for each formulation. We have addressed the suggested comment in the revised manuscript.

-Table 2- Did you use different Lupin/Wheat cultivar for each formulation? Or it was the same? Please, mention this in methodology. In both cases information about Lupin/Wheat needs to be in separate table (maybe in Appendix, since it is not you main objective)

Thank you for your feedback, yes, they are the same cultivars for all the formulations.

Rather than to form a new table, we have added the Lupin/Wheat germ information (yellow highlighted) in the revised text for simple understating of readers.